

# The challenge of forecasting impacts of flash floods: test of a simplified hydraulic approach and validation based on insurance claim data

Guillaume Le Bihan[1], Olivier Payrastre[1], Eric Gaume[1], David Moncoulon[2], and Frederic Pons[3]

[1]LUNAM Université, Ifsttar, Département GERS, Lab. Eau Environnement, Route de Bouaye, CS4, 44844 Bouguenais cedex, France
[2]CCR, 157 boulevard Haussmann 75008 Paris, France
[3]CEREMA DTer Méditerranée, Pole d'Activités Les Milles, Avenue Albert Einstein, CS 70499 Aix en Provence Cedex 3, France

*Correspondence to:* O.Payrastre (olivier.payrastre@ifsttar.fr)

**Abstract.** Up to now, flash flood monitoring and forecasting systems, based on rainfall radar measurements and distributed rainfall-runoff models, generally aimed at estimating flood magnitudes - typically discharges or return periods - at selected river cross-sections. The approach presented here goes one step ahead by proposing an integrated forecasting chain for the direct assessment of flash flood possible impacts on inhabited areas (number of buildings at risk in the presented case studies).

The proposed approach includes, in addition to a distributed rainfall-runoff model, an automatic hydraulic method suited for the computation of flood extent maps on a dense river network and over large territories. The resulting catalogue of flood extent maps is then combined with land use data to build a flood impact curve for each considered river reach: i.e. number of inundated buildings versus discharge. Theses curves are finally used to compute estimated impacts based on forecasted discharges. The approach has been extensively tested in the regions of Alès and Draguignan, located in the South of France,

where well documented major flash floods recently occurred. The article presents two types of validation results. First, the automatically computed flood extent maps and corresponding water levels are tested against rating curves at available river gauging stations as well as against local reference or observed flood extent maps. Second, a rich and comprehensive insurance claim database is used to evaluate the relevance of the estimated impacts for some recent major floods.

## 1 Introduction

Hydro-meteorological forecasts are essential for an efficient real-time flood management, especially when the situation is evolving rapidly. Forecasts provide crucial information to crisis managers for the anticipation and appraisal of the forthcoming floods that may affect areas at risk. In the particular case of flash floods, often affecting simultaneously a large number of small ungauged streams, suitable forecasting systems are still currently under development over the world. The first proposed methods were focused at gauged stream sections, where rainfall-runoff models could be calibrated (Georgakakos, 2006; Norbiato

et al., 2008), leaving a large part of the potentially affected stream network uncovered. The most recent developments aimed at providing forecasts also at ungauged locations. These forecasts rely on highly distributed hydrological models and on radar



based quantitative precipitation estimates or nowcasts (Cole and Moore, 2009; Rozalis et al., 2010; Wang et al., 2011; Javelle et al., 2014; Gourley et al., 2014; Naulin et al., 2013; Versini et al., 2014; Gourley et al., 2017). Such models provide indications of possible flood magnitudes, but are still rarely designed to directly evaluate the possible associated impacts. A large number of simultaneous alarms may be generated in case of a significant rainfall event by such highly distributed flash flood forecasting

systems. And it is now recognized that end-users, such as emergency managers, who have little time for situation analysis and decision making during flash floods, crucially need rapid assessment of the possible field consequences and damage severity (Schroeder et al., 2016; Cole et al., 2016). Moreover, a direct forecast of possible field consequences opens the possibility for assessing the performance of flash flood forecasting systems in ungauged areas, based on reported consequences, as surrogate for measured flood discharges (Versini et al., 2010a; Naulin et al., 2013; Javelle et al., 2014; Moncoulon et al., 2014; Saint-

Martin et al., 2016; Le Bihan et al., 2016). In the near future, real-time assimilation of proxy data for flood magnitude such as information contained in reports of rescue services or social networks could be envisaged. This article presents a proposal of such an integrated flash flood impact forecasting chain and illustrates its validation against insurance claims. If successful, such an approach may be of great help for both crisis managers to better appraise the expected flash flood impacts, and for hydrologists to improve their modeling approaches in ungauged situations.

Translating discharges into local possible impacts requires an estimation of the corresponding flood extent, as well as the knowledge of the level of exposure (location) of the considered assets and possibly of the vulnerability of these assets. This information may be difficult to assess and incorporate at the large scale at which flash flood forecasting systems are implemented to monitor a large number of small river reaches. Large-scale flood mapping approaches based on digital terrain models (DTM) were recently proposed and tested (Yamazaki et al., 2011; Pappenberger et al., 2012; Sampson et al., 2015). These works may

offer an interesting way for an automatic treatment of DTM and flood mapping. But they were not designed up to now for the simulation of a large range of flood magnitudes and were generally applied at relatively large spatial resolutions: computation square grids from 100 metres up to 1 kilometre. Such resolutions are not suited to the representation of floodplains of small streams. On the other hand, detailed flood inundation mapping approaches are available at higher resolutions (Bradbrook et al., 2005; Sanders, 2007; Nguyen et al., 2015), but require large computational resources which limit the implementation possibil-

ity at a large scale. In both cases, most of the proposed mapping approaches would not be compatible with an application in real time.

The approach proposed hereafter combines an applicability at a large scale (computational efficiency), the possibility to be integrated in a real-time forecasting chain and a high spatial resolution for an appropriate representation of floodplains of small ungauged streams. It is proposed to compute automatically (i.e. without manual corrections) a series of flood extent maps for

each river reach, covering a large spectrum of discharge values (i.e. discharge return period values). A DTM treatment method is proposed for the extraction of river cross-sections, that are used in one-dimensional (1D) steady-state hydraulic numerical models for the computation of water stages and flood extent maps. Land use databases are then analysed to compute the number of buildings in the estimated flooded areas for each discharge value and each river reach. Based on this preliminary analysis, a relation between the discharge and the number of affected buildings is adjusted at the river reach scale and is then used as

impact model of an integrated rainfall-runoff-impact simulation chain.





Even if the proposed procedure may appear relatively straightforward, the main challenge lies in its automatic application and validation over extended territories on a dense stream network - typically streams with upstream watershed areas larger than $5\ km^2$. The validation is an essential step which should reveal if such a forecasting chain is able to provide reasonably accurate results, despite the necessary simplifications of such large scale applications (standard roughness coefficient values, 1D steady-state hydraulic models, missing bathymetric data,..), and sources of uncertainty (DTM accuracy, unknown vulnerabilities,..).

This article presents the proposed method and its application on two well-documented test case studies. Two types of evaluations are conducted. First, the automatically computed flood maps and corresponding water levels are tested against the rating curves at river gauging stations as well as against local reference or observed flood maps. Second, a rich and comprehensive insurance claim database provided by the main French reinsurance company (Caisse Centrale de Réassurance) is used to evaluate the relevance of the estimated impacts (number of possibly inundated buildings) for some recent major floods. The article is organized as follows: the first section presents the methodology developed for both, the implementation of the impacts model, including the computation of a catalogue of flood extent maps, and for the implementation of the rainfall-runoff-impacts simulation chain; section 3 presents the two application case studies and the datasets used for the validation; section 4 exposes and discusses the obtained results.

## 2 The proposed rainfall-runoff-impact simulation chain

### 2.1 Simplified automatic implementation of 1-D steady-state hydraulic models

The Cartino method (Pons et al., 2014), has recently been proposed to automatically build the input files and run one-dimensional (1-D) hydraulic models based on data extracted from high resolution Digital Terrain Models (DTM). This method has been used and adapted herein to derive catalogues of flood extent maps for a wide range of discharge values. The software proceeds in three steps (figure 1). First, the locations of the cross-sections are selected and their shapes extracted from the DTM. Second, the corresponding input files are built and the selected 1-D hydraulic model is run to compute longitudinal water level profiles corresponding to each selected discharge value: the Mascaret 1D model (Goutal et al., 2012) has been used in the present case study. Third, a post-treatment intersects the water level profiles and the DTM to compute the flood extent and water depth maps.

To ensure an automatic computation, important simplifications are introduced in the structure of the hydraulic model: cross-section shapes are estimated based on a simple extraction from the available DTM, without additional information on topography or bathymetry; specific sections such as weirs or bridges are not represented; a unique roughness coefficient is used for all stream reaches (n=0,05 hereafter); no distinction is made between river bed and floodplain. Of course all these assumptions, even if necessary for sake of simplicity, may have an impact on the accuracy of the results. This point will be evaluated and discussed in the next sections.

The first two steps are run in an iterative way to adjust the width of the cross-sections and their inter-distances for each considered discharge value. The cross-sections should be wide enough to contain the simulated flow and successive cross-sections should not overlap. The procedure is initiated based on a first estimation of the possible extent of the flooded area



(provided as input as well as the position of the river bed), which is used to define the initial width of each section. The distances between profiles are then defined as a proportion of each cross-section width (proportion defined as input parameter). After each run, it is checked if the computed water level does not exceed the altitude of the borders of the cross-section. If it is the case, the cross-section is enlarged in a proportion also defined as input parameter. Distances between profiles are adapted in

consequence. Note that the final set of cross-sectional profiles and their location vary between the runs and depend in particular on the considered discharge value.

The choice of the input information (default extent of flooded areas and parameter values) influences the result and the computation time. For instance, a too wide initial flooded area or a too fast increase of cross-sections widths may lead to incorporate depressions of the flood plain (perched river bed) that are not connected to the river bed and hence not active for

the considered discharge (see figures 1.b and 1.c). This may lead to both, overestimate the local extent of the flooded area and to decrease the simulated water level, with an impact on the upstream cross-sections due to backwater effects. On the other hand, a too narrow initial flooded area and/or too slow increase of the widths of the profiles highly increases the computation time. To cope with these difficulties, the method offers the opportunity to modify manually any profile after a first iterative run.

## 2.2 Computation of a catalogue of flood extent maps

The hydraulic computations are conducted for each river reach of the considered stream network, and for a range of discharge values corresponding to return periods of 2, 5, 10, 20, 50, 100, 200, 500, and 1000 years. The downstream limit condition for each reach is the water level computed for the downstream river reach for the discharge of the same return period. The discharge quantiles are estimated based on the French SHYREG flood frequency database (Aubert et al., 2014). Nevertheless, the accuracy of this information on flow frequency is not crucial for the implementation of the impact model: it just enables to

derive flood maps for defined discharges values of relatively homogeneous magnitude for all the considered river reaches.

An automatic check is performed after each hydraulic computation to eliminate the main errors in the shapes of cross-sections, mainly associated with the limits of DTM information used as input: bridges still appearing in the DTM, remaining noise due to dense vegetation. This verification is based on the comparison between the wetted areas of the successive cross-sections: automatic removal of cross-sections appearing as inconsistent with the immediate upstream cross-section (ratio

between successive wetted areas exceeding 3), before running again the hydraulic model.

One crucial aspect for the computation is the delimitation of the active river bed in cases of perched rivers. The computations are conducted step by step to limit the risk of including too rapidly flood plain depressions in the cross sections. For this purpose, the computations are first conducted for the smallest discharge (2-year flood), using a narrow initial extent of flooded area as input of the Cartino method. The width of the cross-sections are then progressively increased to detect as accurately as possible

the limits of the river bed. The computed flooded area is then used as initial cross-section extent for the the next computation (next larger discharge value), and so on.

In addition, a post-treatment procedure is applied to ensure a better overall consistency of the catalogue of flood extent maps. Computed flooded areas disconnected from the river bed are systematically removed form the final result (see figure 1d). And



computed flood extent maps are corrected to systematically include the flood extent maps computed for the lower discharge values.

Despite all aforementioned precautions, some sources of errors affecting the quality of results remain. These errors are mainly related to the simplification of the procedure necessary for a fast and automatic implementation over a large river network. These errors are mainly due to:

- errors in the retrieval of the shapes of the river cross-sections due to the automatic extraction and to the limits of topographic information used (DTM),

- absence of representation of friction losses due to bridges and other hydraulic singularities,

- choice of a fixed Manning roughness coefficient, equal to 0.05,

- steady-state regime computations,

- remaining difficulties to determine the active river cross-section in cases of perched river bed despite the previously described precautions.

For these reasons, the information obtained should not be considered as a highly accurate estimation of flooded areas, but as an estimation giving an order of magnitude of the level of flooding and enabling some comparisons at a regional scale.

## 2.3 From flood extent maps to impact assessment

Based on the catalogue of flood extent maps, an impact model is derived for each river reach. The hillslope limits of the rainfall-runoff model are used to delineate the river reaches and the corresponding flooded areas (figure 2.a). The assets present in the estimated flooded areas are counted. The number of buildings may be used as an indicator of human asset exposure. More precisely, the number of private houses with a single and geo-referenced insurance policy in the available insurance database has been considered herein to enable comparisons with the reported number of insurance claims (see section 3.6). The values computed for the 10 flood extent maps for each river reach are then linearly interpolated to build a continuous relation between discharge and number of flooded houses, i.e. number of impacted insurance policies (see figure 2.b).

## 2.4 The rainfall-runoff-impacts simulation chain

The impact model is finally incorporated in a full simulation chain combining radar-based quantitative precipitation estimates as input (see section 3) and a distributed rainfall-runoff model for the simulation of discharges over the stream networks. Quantitative precipitation forecasts or nowcasts may also be used as input of the chain to increase anticipation lead times. However, quantitative precipitation estimates have been considered herein to focus the analysis on the accuracy of the proposed rainfall-runoff-impacts simulation chain.

The CINECAR hydrological model (Gaume et al., 2004; Naulin et al., 2013; Versini et al., 2010b) was selected to build the simulation chain. CINECAR is a distributed rainfall-runoff model based on a representation of the catchment as a ramified



series of stream reaches, to which both left and right-hand hillslopes are connected. The spatial resolution of the model (areas of hillslopes and associated length of river reaches) can easily be adapted and has been defined herein to match the resolution of the impact model.

CINECAR only simulates the rapid component of the runoff and is suited for modelling the rising limb and peak phases of significant flash-floods. For sake of simplicity, the hillslopes are represented by schematic rectangles of the same area as the actual hillslopes and the river reaches are assumed to have a rectangular cross-section. The width of the cross-section varies with return period of the discharge. The Soil Conservation Service-Curve Number model (SCS-CN) is used to compute runoff rates and the corresponding effective rainfall at each computation time step (USDA, 1986). The effective rainfall is then propagated onto both the hillslopes and the river network using either the kinematic wave model (Borah et al., 1980), or the Hayami solution for diffusive wave model (Moussa, 1996) to represent the flood wave attenuation in river reaches with slopes lower than $0.6\%$.

Since CINECAR was developed for the purpose of forecasting flood hydrographs in ungauged catchments, it includes a limited number of calibration parameters (Naulin et al., 2013): the width of river reaches is the main parameter controlling the transfer function and is estimated based on the Strahler order of river reaches; the Curve Number (CN) value is the second key parameter which controls the temporal evolution of runoff rates. The USDA method is used for the estimation of the CN (USDA, 1986), depending on the bedrock type, land use and antecedent rainfall. This model was applied in 2013 to the entire Cévennes Region (see section 3), and validated with respect to measured data (Naulin et al., 2013). It provided satisfactory results for large flood events, similar to the ones obtained with locally calibrated standard conceptual rainfall-runoff models, with an average Nash criterion computed for single flood events equal to 0.49.

The same model version has been used herein without any further adjustments for the Argens watershed (second case study), which characteristics are relatively similar to the watersheds of the Cévennes area. The performances of the model on this new application case study could be verified according to stream gauge and peak discharge data available for the June 2010 event.

The discharge values computed with the CINECAR model are finally converted into estimated impacts according to the continuous discharge-impacts relations defined for each river reach.

## 3 Presentation of the two case studies

### 3.1 The region of Alès in the Cévennes area, south eastern France, and the September 2002 flood

The region of Alès is located in the core of the Cévennes region, well known to be prone to frequent and intense flash floods (Gaume et al., 2009). Moreover, it has been identified during the implementation of the European Union (EU) flood directive as one of the areas the most exposed to flood risk in France (Areas with Potential Significant Flood Risk -APSFR- selected for the implementation of risk management plans). The region is presented in figure 3, indicating the exact limits of the APSFR of Alès. Its vulnerability to floods is mainly related to the presence of the town of Alès, but also to other highly vulnerable smaller towns such as Anduze (see figure 3).





This territory is part of two main watersheds: the Gardon d'Alès and the Cèze rivers. These two main rivers have their upstream course in the Cévennes relief, and reach in their downstream part a plateau area with limited slopes. The APSFR of Alès is located in the transition zone between the mountainous and plateau areas. Therefore, this case study includes a large variety of river bed configurations including steep and narrow v-shaped valleys as well as flat and wide floodplains. Some statistics about the river bed characteristics are provided in table 1.

The region was hit on September 2002 by a catastrophic flash-flood event (Delrieu et al., 2005). 23 casualties and 1.2 billion euros of damages were reported. A maximum rainfall accumulation of 680 mm in 24 hours was recorded in the town of Anduze. This event induced a large number of insurance claims. It has therefore been selected herein for the evaluation of the impact simulation chain.

The area selected for the study corresponds to the exact limits of the APSFR of Alès (area of about 1000 $km^2$). It includes 400 km of river streams having at least a 5 $km^2$ upstream catchment area. The stream network has been divided into 192 river reaches (a river reach being defined as the portion of river located between two confluences), among which only 70 reaches (132 km) are covered by the current operational flood forecasting service (www.vigicrues.gouv.fr: main network on figure 3).

### 3.2 The region of Draguignan in the Argens watershed, south eastern France, and the June 2010 flood

The Argens river watershed (2700 $km^2$) is located in the eastern part of the French Mediterranean region. It has been hit by several severe flash floods in the recent years. This watershed was also selected as an APSFR for the implementation of the EU flood directive.

Among recent flash floods observed in this region, the June 2010 event is certainly the most catastrophic one. It particularly affected the region of Draguignan located in the eastern part of the Argens watershed. 25 casualties and 1 billion euros of estimated damages were reported after this event as well as a large number of insurance claims.

The region of Draguignan is presented on figure 3. As for the Alès case study, it presents a varied topography, and consequently a wide variety of river bed configurations, from narrow valleys in the upstream part of the studied watersheds, to wide floodplains in their downstream part. The area includes 345 km of rivers with at least 5 $km^2$ upstream catchment areas, which were divided into 173 river reaches. Only 42 of these river reaches are covered by the operational flood forecasting service. This illustrates the possible added value of the proposed integrated flash flood impact simulation chain in France.

### 3.3 Available Digital Terrain Models

The implementation of the impacts models is based on commonly available high resolution DTMs in both case studies. Nevertheless, the characteristics of these DTMs significantly differ:

- a 20 m resolution DTM (interpolated at 5 m) produced in 2007 by the Conseil Général du Gard was available in the case of the Alès case study; its altimetric accuracy was estimated to less than 20 cm in non-vegetated areas, and less than 1 m in vegetated areas,



- a 5 m resolution DTM extracted from the IGN RGE Alti database was available in the case of the Draguignan region; its altmetric precision ranges from 20 cm (main rivers covered with lidar data) to 70 cm (other areas covered with photogrammetry products for instance), and artificial structures such as bridges are removed in this DTM; this second product can be considered as standard DTM data that will be available over the whole French territory by the end of 2017.

It should finally be noted that if the terrain information used herein is now commonly available in France, its accuracy remains limited and may affect the quality of results: DTMs extracted from Lidar measurements are currently limited to the main rivers which are not the scope of this study, and would probably lead to results of better quality. The geographic coverage of Lidar data is however evolving very fast, and this data should become available also for small rivers in a near future.

## 3.4 Rainfall and discharge data

Both regions are equipped with relatively dense stream gauge and rain gauge networks, complemented with weather radars. In the case of the APSFR of Alès, the whole dataset was carefully checked in the framework of the OHM-CV research observatory (www.ohmcv.fr). It can therefore be considered as exceeding conventional quality standards.

The locations of stream gauges are shown on figure 3. Given the limited possibilities to conduct direct flow measurements during intense flash floods, the rating curves are often extrapolated, with consequently a reduced accuracy of estimated discharges for high water levels and large floods.

Radar based QPEs are available for both case studies. They correspond to the operational Meteo France Panthere QPEs in the case of the Draguignan June 2010 event, and to a radar QPE reanalysis provided by the OHM-CV research observatory in the case of the September 2002 event in the region of Alès (Delrieu et al., 2009).

## 3.5 Reference flood maps

Thanks to the recent application of the EU flood directive in both considered areas, a great effort was put on mapping flooded areas, leading to detailed inundation maps available for three reference events: the 30 year return period flood ("common" event), 300 year return period flood ("medium" event), and 1000 year return period flood ("large" event). These maps were generally obtained based on an 1-D hydraulic modeling, carefully implemented by experts in hydraulics. These maps were criticized and validated using all available information, including the observed extents of inundation during past floods such as the September 2002 and June 2010 floods. These maps were used as a reference here for the evaluation of the computed catalogues of flooded areas. Unfortunately these maps were produced on only part of the considered river networks. This limits the validation possibilities. In the case of the Alès case study for instance, the river network covered by reference maps represents 192 km (out of 400 km included in the case study) and includes 84 river reaches (out of 192).





### 3.6 Insurance claim database

During the last 15 years, an insurance claim database has been created and supplied by the Caisse Centrale de Réassurance (CCR) within the framework of its reinsurance contracts with its clients (Moncoulon et al., 2014). This database covers the whole French territory, and the quality of data is considered as acceptable for the period since 1997. It includes information on

both nature and location of insurance policies and claims for all events classified as so-called "CATNAT" events. "CATNAT" are flood events with an estimated return period exceeding 10-years and consequently for which the natural disaster insurance compensation is activated, in accordance with the compensation scheme implemented since 1982 in France.

The CCR database is certainly the most comprehensive available database on flood field consequences existing in France. It has nevertheless some limits, and the content and accuracy of the insurance policy and claim data incorporated have evolved

over the years. Consequently the data had to be carefully selected to enable an objective comparison with the modeling results:

– With regard to the policies, the database nowadays includes more than 80% of the policies of the French insurance market, insurance against natural hazards being mandatory in France. A great effort has also been put on the accurate geo-referencing of the policies in the recent years: approximatively 70% of the policies are geo-coded at the street number, about 15% geo-coded at the street center, and 14% geo-coded at the town centre. Only the policies accurately

located (i.e. geo-referenced at the street number or center) could be considered here for incorporation in the impact model. Note that the address of the policy is generally the address of the owner of the asset. It does often, but not always, correspond to the address of the insured asset. Note also that in case of flats, several policies can be located at the same address, but are generally not all exposed to flooding. Therefore, only insurance policies and claims corresponding to private houses were considered here to ensure a more direct correspondence between houses, exposed policies and

claims.

– As far as the claims are concerned, their collation in the database is less systematic: between 30 and 50 % of the total number of CATNAT insurance claims for the French market are documented in the database depending on the year. This depends on the comprehensiveness of the data provided by the insurance companies to the CCR. To base the comparison on faithful and robust data, the policies of the insurance companies documenting more than 80% of their claims in the

CCR database were selected here for the validation of the forecasting chain. The claims with a null compensation amount were also removed, since there is no certainty in this case that a real damage occurred: at least, the amount of damage did not exceed the insurance excess in these cases.

Finally, only part of the database could be considered for the comparisons: geo-referenced policies, private houses, comprehensiveness of the information on claims provided by the insurance companies. A comparison with the IGN BD Topo database

(buildings of < 7m height corresponding mostly to individual houses) shows that the number of selected policies represents about 20% of the total number of buildings in the considered floodplains (table 2). The proportion is higher in the Draguignan 2010 case, sign of an improved quality of the CCR database since 2002. To enable a direct comparison of the forecasted impacts with the number of reported claims, the impacts model (figure 2) was finally built based on the selected policies: the forecasted



information corresponds in this case to a number of possibly impacted individual houses, for which almost comprehensive claim data is available. The details of the insurance data of individuals being confidential, the validation was based on claim data aggregated at the river reach scale. Moreover, to ensure a total confidentiality, the analyses and comparisons were only conducted on river reaches with at least 20 recorded policies in the database.

The analysis of the available claim data reveals some additional surprises. First, despite the comprehensiveness of the selected claim data, the ratio between reported claims and policies does rarely exceed 50% even in areas which are likely to have been flooded (figure 7). Some houses with raised basements may be out of water even in the flooded areas, but not in such a high proportion. This ratio is also explained by the remaining proportion of claims not documented in the database (up to 20% according to the data selection), by the significant proportion of claims with a null compensation amount, and maybe also by

some non-declaration of flood damages to the insurance companies. The large difference between the number of policies and the number of reported claims is a common feature for all floods in the CCR database (Moncoulon et al., 2014). In total, the combination of the limited proportion of selected insurance policies (policy/building ratio) and of the partial documentation of the claims and damages (claim/policy ratio) leads to a relatively low average ratio between the number of buildings and the number of reported claims in the floodplains: 6 to 9% (table 2). This explains why, despite the richness of the CCR database,

the evaluation of the proposed forecasting chain based on insurance claims, with the ambition to provide results at the stream reach level, had to be limited to extreme flood events with large numbers of reported claims.

The second major surprise is the significant proportion of reported claims located outside the estimated 1000-year flood envelop (over 10% for the two considered events, see figure 7). This is also a general feature observed in the CCR database. Several explanations can be put forward to explain the presence of claims outside the identified flood areas: (i) damages may

be induced by small watercourses not represented in the flood model, (ii) they may also be triggered by other local processes (runoff accumulation in low points, sewers saturation, cellar flooding due to groundwater raising, ...), and (iii) the address of the owners of the insurance policies may not correspond to the location of the affected assets. No information in the database enables a distinction between damages induced by direct stream flooding and other processes. The existence of a significant proportion of claims not directly related to river overflows represented by the model adds to the complexity of the validation

exercise. For the purpose of this validation, and considering that the contours of the flooded area are only available for a limited number of the considered river reaches, every claim located in the maximum possible flood envelop (estimated 1000-year flood envelop) for each stream reach has been considered for the computation of the reference number of observed claims per reach. This could lead to overestimate the reference number of claims for reaches with actual limited flooding and will be discussed in the section presenting the results.

Despite all these constraints which limit the information content of the data, this CCR database is still a rich and unique source of information to measure the impacts of flash-floods in small rivers. It has been until now been used to assess economic losses at the event scale (Moncoulon et al., 2014). It was worse testing if it could provide a number of private houses affected by the floods for each river reach to be compared to the outputs of the proposed forecasting chain.



## 4   Results and discussion

As mentioned in the introduction, the results are presented hereafter in two steps:

- first, an evaluation of the accuracy of the catalogue of flooded areas is presented based on the Alès case study. Two different types of evaluations results are exposed: the water levels estimated at stream gauges are compared to existing stage-discharge relations (rating curves), and the estimated flooded areas are compared to reference areas computed for the purpose of the implementation of the EU flood directive.

- in a second step, the results of the whole rainfall-runoff-impacts simulation chain are presented for the both case studies, respectively for the september 2002 and june 2010 floods, and evaluated against real observed flooded areas and insurance claim data.

### 4.1   Comparison of water levels at stream gauges locations (Alès case study)

A first evaluation of the results of hydraulic computations is proposed here based on information available at stream gauges. These gauges indeed offer locally the opportunity to compare the rating curves based on expert know-how with the results of the 10 steady state hydraulic computations used for the implementation of the impacts model. Considering that the distance between cross-sections may reach up to 100 meters in the proposed method and that their locations is variable, additional cross-sections corresponding to the exact locations of the stream gauges were manually added for the hydraulic computations to enable comparisons.

The results are presented on figure 4 for three different stream gauges. This figure illustrates contrasted situations, which are detailed hereafter.

The case of the Mialet station (figure 4.a) appears as an ideal situation, where the shape of the cross-section is well retrieved from the DTM and the computed water levels are very close to the existing rating curve. Note that the real measured discharge values are low for the three stream gauges. The comparisons are focussed on the extrapolation range of the rating curve: i.e. the automatically implemented hydraulic model is essentially compared with local expert know-how generally based on a detailed hydraulic computation. The very satisfactory result obtained in Mialet may be explained by the simple shape of the cross-section (deep and relatively narrow in this case), the limited presence of vegetation in the river bed that could affect local roughness, and the significant slope (i.e. limited risk of backwater influences). The selected roughness coefficient value of 0.05, an average value based on post-event studies (Lumbroso and Gaume, 2012), corresponds well to the locally adjusted one.

The comparison for the Banne station is less satisfactory (figure 4.b). The two cross-sections have similar shapes but do not seem to have the same reference altitude. A difference of about two meters exists for an unknown reason. This may nevertheless have little influence on the relative water levels and corresponding computed flooded areas. But, if the computed water levels are reduced by two meters, the computed discharges still appear much larger than the corresponding discharge estimates based on the local rating curve for the larger stage or discharge values. The slope of the local rating curve appears very low and does not even follow the evolutions of cross-sectional areas with the water stages. Such a rating curve shape could result form a local backwater effect and could illustrate the limits of the hydraulic model used, that does not account for such phenomena. In





this case, there is nevertheless no hydraulic singularity immediately downstream the gauge that could generate such an effect. The reference extrapolated rating curve is questionable.

The case of the Alès station illustrates other sources of difficulties (figure 4.c). Again in this case, the topography of the river bed appears well retrieved from the DTM even if a horizontal displacement is noticeable. But the shape of the computed

stage-discharge relation, even if on average close to the rating curve of the station, appears chaotic and non monotonous. The stream gauge is located just upstream a large meandering where the valley is perched with a large flood plain on the right bank (figure 4d). This flood plain is only inundated during extreme floods. Depending on the run (i.e. on the discharge value), it is included or not in the modelled cross-sections located just downstream the gauge and may generate an inundated area not connected to the river bed, finally eliminated in the post-treatment as described previously. But this artificial inclusion of

the flood plain in the cross-sections, for some intermediate discharge values, leads to under-estimate the computed water level at the gauged cross-section due to the backwater propagation (see longitudinal profiles in figure 4.d). Clearly, the proposed procedure could not eliminate all the problems encountered when modelling perched rivers with 1-D hydraulic models, despite the precautions taken. The use of a 2-D hydraulic models could help solving the problems encountered in the future but at the price of a large increase of computation times. A detailed analysis of the results obtained for the two case studies nevertheless

reveals that the number of remaining problematic river reaches is limited. Moreover, these problematic configurations mainly correspond to relatively large rivers ($315 km^2$ of drainage area at the Alès station), which correspond to the limit of the target application domain of the proposed method.

Similar results were obtained for the other available gauging stations. Overall, the results are extremely satisfactory, almost exceeding the initial expectancies. The cross-sectional shapes can be correctly retrieved from the existing DTM despite their

limits, at least sufficiently accurately for the reconstruction of local stage-discharge relations. It is important to note that the summer flow of these Mediterranean rivers is reduced. This explains why bathymetric data is not really needed. The selected average roughness coefficient value appears to be suited to the local expert know-how. In the future, checking stage-discharge relations at gauging station could help to adjust the values of roughness coefficients in the proposed approach extrapolated to other areas. It is important nevertheless to keep in mind that hydraulic singularities such as bridges, that can not be characterized

through the DTM, may locally largely influence the stage-discharge relation, even if this could not really be illustrated on the presented case study.

### 4.2 Comparison with reference flooded areas (Alès case study)

Reference flood maps have been produced for the purpose of the implementation of the EU flood directive for discharges values corresponding to the 30-year and 300-years flood events for almost half of the considered stream reaches. The automatically

computed maps could be compared to these reference maps for the same discharge values. For each river reach, the estimated surface ($ES$) and observed or reference surface ($RS$) are compared. The surface in common ($S_c$) as well the excess surface ($S_e$: computed but not observed) and default surface ($S_d$: observed but not computed) are evaluated (see figure 5a). Note that $ES = S_c + S_e$ and $RS = S_c + S_d$. A synthetic incoherent surface ratio (ISR) is computed (eq. 1):





$$ISR = \frac{S_e + S_d}{RS} \qquad (1)$$

The permanent river bed (represented on figure 5a) which is not affected by estimation uncertainties, is not considered in the computation of the surfaces $ES$ and $RS$ and hence in the computation of $ISR$.

Figure 5.b presents the distributions of ISRs computed for the 84 river reaches for which reference inundation maps were available (71 reaches for the 30-year flood) for the Alès case study. The results appear overall satisfactory: the IRS ratio rarely exceeds 30 %. This ratio includes both default and excess surfaces: the real difference between $ES$ and $RS$ is in fact more limited. This suggests that the errors in the estimation of impacts will also be more limited. The IRS ratio is sensitive to small differences between computed and reference maps as illustrated by the examples shown in figure 6 and low values are difficult to reach and suggest a quasi perfect agreement. Figure 5c also shows that the ISR values depend on the magnitude of the simulated floods. The results obtained for the 300-year flood appear much more accurate, with ISRs almost never exceeding 50 %. This can be explained by the fact that the floodplains are largely flooded in case of high return period discharges, with limited possibilities for large errors. The estimation of the flooded areas for the 30-year discharges appear less accurate, with a significant proportion of stream reaches with large relative errors: ISRs exceeding 100 % for almost 10 % of the reaches. These relative errors are nevertheless essentially related to observed flooded surfaces lower than 0.1 $km^2$ (figure 5.c). This corresponds to the very beginning of the river overflow that is hardly captured accurately with a hydraulic model based on automatic extractions from a DTM, whose roughness parameters are averaged over large areas and with no description of local hydraulic singularities. But large relative errors on small flooded areas will have limited influence on the impact model. Overall, the simulated flooded areas correspond pretty well to the reference areas. These results confirm the validation results obtained on the rating curves: the proposed 1-D hydraulic model, based on automatic extractions from relatively accurate DTM and on a regionally averaged roughness coefficient, stands overall reasonably well comparison with local expert-based hydraulic models, at least in the test region. Some reasons can be put forward to explain this result: the limited need of bathymetric data in the Mediterranean context ; the river slopes - typically 0.5% or more - limiting the distance over which backwater effects propagate and therefore the local influence of hydraulic singularities not taken into account in the model ; the reduced sensitivity of water stages ($h$) to variations of discharges ($Q$) due to the usual shape of stage-discharge relations ($h \propto Q^{3/5}$) ; and finally, the reduced sensitivity of the flooded area to the water stages, except at the beginning of the river overflow, due to the cross-sectional shapes of the river beds, with generally narrow valleys and well delimited flat flood plains.

### 4.3 Accuracy of forecasted flood extents for the Alès 2002 and Draguignan 2010 floods

For both considered validation flood events, observed inundated areas were carefully mapped after the floods on part of the affected streams. These observed flood extent maps were compared for each river reach to the forecasted flood extents. More precisely, the incoherent surface ratios (ISR) described in the previous section were computed based on the map of the flood catalogue corresponding to a discharge value immediately lower or equal to the forecasted peak discharge value. There is indeed not necessarily a map in the catalogue corresponding to the exact value of the forecasted peak discharge. This choice should





lead to a slight underestimation of the flooded areas if the hydrological and hydraulic models were perfect. This is illustrated in figure 6 which shows for some river reaches the observed flood extent and both extents corresponding the discharge values immediately lower and higher to the forecasted peak discharge. This figure also shows to what range of differences the ISR values correspond.

The ISR ratios obtained for both events are summarized in figure 5.d. Without surprise, the ISR values are significantly increased when actual flood events are considered, if compared to the initial evaluation of the catalogue of flooded areas presented in the previous section. When actual floods are simulated, additional sources of uncertainties affect the computed flooded areas. The simulated peak discharge on which the forecasted maps are based may differ significantly from the observed ones. Moreover, the observed flooded areas may be the result of local processes (dike breaches, blockages) particularly during

extreme flood events. These processes are not represented in the hydraulic models - the proposed simplified regional model or the local models used to elaborate the reference flood maps. In fact, figure 6 shows that the differences between observed and simulated flood extents are not only explained by uncertainties in the forecasted peak discharge values which would result in systematic over or under-estimations of flood extents, but also by local processes imperfectly accounted for in the hydraulic models.

Finally, a large proportion of the relative ISR remain lower than 50%. Consequently, the computed flood extent maps may be sufficiently realistic to provide an approximation of the local field consequences of floods and of their spatial distribution. This is verified in the next section. It is noteworthy that the ISR are higher in the case of the Draguignan 2010 flood even if the models (rainfall-runoff and hydraulic models) have been extrapolated to this event and area without any further calibration. It could be explained by the higher accuracy of the DTM available in this area. In any case, it is a promising results that seems

to reveal that uncertainties related to the extrapolation of the models may be of secondary importance if compared to the other sources of uncertainties of the proposed approach. General conclusions can nevertheless not be drawn from one single example.

### 4.4    Validation of forecasted impacts based on insurance claims

Even after a careful selection of the appropriate validation data, the observed claim/policy ratio is significantly lower than 1 (table 2 and figure 7), and varies between river reaches. As exposed in section 3.6, this is explained by claims with no

compensation and/or non-declared claims, buildings with raised basements, inaccurate location of insured buildings, and also the imperfect filling of the claim database. Clearly, the number of reported claims per river reach has a random component. To account for randomness in the validation process, the number of reported claims can for instance be considered as the result of a random binomial process $B(n,p)$, $n$ being the number of policies in the considered flooded area and $p$ the probability that a corresponding claim with non-null compensation is observed. If $p$ is considered to be the same for all reaches and equal to

the average claim/policy ratio for the selected sample of insurance policies, a confidence interval (90% binomial confidence interval herein) can be estimated for the number of claims corresponding to every computed number of impacted policies (policies located in the estimated flooded area). The result is presented on figure 8 for the two case studies.

This figure shows a relatively good agreement between the forecasted number of impacted policies and the number of reported claims per river reach. The observed spread of the results nevertheless exceeds the width of the 90% intervals, especially



in the Draguignan 2010 case study: more than 10% of the dots lie outside the confidence limits with predominantly underestimations by the model. This indicates that some other sources of errors affect the relation between forecasted and observed number of claims. First, the reported claim versus policy ratio is significantly affected by water depth and flood duration, two variables that have not been considered herein. Second, the reference number of claims has been computed based on the maxi-

mum possible estimated flood envelop (1000-year flood envelop) to account for possible errors in the computed flooded areas. Since a significant proportion of claims are not related to the streams represented in the model (see section 3.6), this choice may lead to overestimate the reference claim number for stream reaches where limited overflow occurs, especially in densely urbanized areas (typically points A and B on figure 8.b). The number of reported claims in the computed flooded areas has also been estimated and the difference with the reference number of claims is indicated by the dotted lines on figure 8. This

difference is modest in the Alès 2002 case, where the floods have been extreme over the entire considered area. The correction is much more significant in the Draguignan case, especially for almost all the points located over the 95% confidence limit (figure 8.b). Several of these reaches, particularly points A and B, are located in the upper South-Eastern part of the studied area, which has not been affected by the most intense rainfalls. The flood extents were only partially mapped in this area after the 2010 flood, but the mapped extent indicate moderate overflow in this sector and are in good agreement with the modeled

flooded areas. The high number of claims located outside areas flooded by the modeled streams and the impossibility to separate these claims, as well as the absence of reference flood extent maps for stream affected by moderate floods clearly limits the use of the insurance claim database for the validation of flood impact models.

Finally, figure 9 compares the spatial distributions of the simulated peak discharges, of the maximum forecasted impacts (i.e. number of flooded private houses with geo-referenced policy) and of the number of associated claims according to the CCR

database. It should be noted that information on claims is provided only for river reaches with at least 20 policies recorded in the database. This explains why no values have been provided for several stream reaches, essentially non-urbanized reaches with limited exposed assets. This figure illustrates the large differences in the outputs of the hydrological rainfall-runoff model (figures 9a and b), and of the integrated rainfall-runoff-impacts modelling chain (figures 9b and c) which provides a much contrasted analysis. It also shows a good overall consistency between the forecasted impacts (figures 9b and c) and observed

claims (figures 9e and f) for the location of the main hotspots. Apart from some exceptions, the ranking and magnitudes of the field impacts appears to be well captured by the proposed forecasting chain. That is mainly the information needed by rescue services to adapt and dispatch their rescue means during flood event management. Such information could also help targeting more effectively alert messages.

## 5   Conclusions

Flood event managers need to assess, in real-time, the severity of possible field consequences associated to hydro-meteorological forecasts, to be able to take appropriate decisions. Automatic assessment methods are necessary in case of fast-evolving events such as flash-floods, when little time is available for information processing and analysis. Moreover, the direct estimation





of field consequences opens the possibility to test the performances of forecasting chains in ungauged areas, where various observations related to damages may be available.

This paper has tested the potential of simple approaches for the estimation of the magnitudes of possible field consequences within flash-flood forecasting chains. The proposed methods have been selected to be implemented with limited calibration
effort, over extended areas and at detailed spatial scales. A particular attention has been paid to the performance evaluation of the proposed chain. An original and particularly rich and comprehensive insurance claim data base has therefore been used. It is to our knowledge the first time an insurance claim data base is used for such a purpose.

The proposed approaches certainly deserve further validation, but the results presented herein on two case studies appear extremely satisfactory and promising. The flood mapping based on 1D steady-state hydraulic modelling, automatically imple-
mented over a large river network with an average roughness value, stands the comparison, for most of the considered river reaches, with local expert-based hydraulic simulations: stage-discharge relations at gauging stations and flood maps computed for the implementation of the European flood directive. The whole hydrological and hydraulic simulation chain provides maximum flooded areas that also appear generally close to the flood extents mapped after the two test events when such maps where available. The typical configuration of the streams affected by flash floods can be put forward to explain such satisfactory re-
sults: (i) narrow valleys and well-delimited flood plains, (ii) limited need for bathymetric data for Mediterranean streams, (iii) steep stream slopes implying limited spatial influence of hydraulic singularities (bridges) and their induced backwater effects, (iv) huge discharge contrasts between moderate, large and extreme floods that are well captured by rainfall-runoff models despite the inevitable modeling uncertainties. The slightly better results obtained in the Draguignan case suggest that some improvements could still be achieved with more accurate topographic data, especially Lidar data enabling a better retrieval of
the cross-sectional shape of the main stream bed. The influence of the DTM accuracy on the results of the proposed approach will have to be tested. Some other difficulties nevertheless remain. The proposed modeling approach has to be further improved to properly handle complex hydraulic configurations such as perched river beds (figure 4). And the observed flood extents may also locally differ from the estimated flooded area even if the discharge value has been well forecasted due to local effects difficult to anticipate such as blockages and breaches. A perfect fit is out of reach. This suggests that we should not put too
much confidence in theoretical flood maps computed either a priori or in real-time. Such maps, if provided to the flood event managers, should be presented as indications of possible flood scenarios close but not identical to the actual flood.

The validation of the estimated damages based on insurance claims faced some difficulties related to the specificity of insurance data. The CCR data base used is probably the most comprehensive source of information about flood insurance losses in France. However, the validation process requires both an accurate geocoding of insurance policies and a comprehensive
information on claims, which limits the amount of available information. A high proportion of claims is also not related to the streams included in the model, limiting the possibility to use this data as a reference for moderate floods if the actual flood extent is unknown (figure 8). The validation exercise could nevertheless be successfully achieved for the two extreme floods studied herein, providing an interesting additional information on the accuracy of the whole simulation chain. It should also be considered that the quality of insurance data is continuously increasing and that some of the limits identified here
(geocoding, comprehensiveness of claim information) should be significantly reduced in the future. Therefore, insurance claim



data should be considered as a relevant option for the validation of flood forecasting results, particularly in the case of flash floods affecting ungauged rivers. Despite this data is generally confidential, is may be accessible through partnerships with insurance companies. Other source of information on flash flood impacts could also be used such has the logs of emergency services, emergency calls, information shared on social networks (USDHS, 2012; Jongman et al., 2015; Tkachenko et al., 2017)

5   or information gathered on the field after or during the event (Ortega et al., 2009; Ruin et al., 2014). This information is also affected by uncertainties and severe biases, especially in flash flood situations: absence of information due to local breakdowns of communication networks, reduction of social network activity and partial filling of emergency logs in strongly affected areas during the paroxysm of the event... Some of this information has nevertheless the advantage to be available in real time (digitized logs, emergency calls, social networks) and could help validating and improving forecasted impacts. Finally, the

10  combined use of flood impact forecasting models and field data mining and processing methods is without doubts a promising avenue for the development of innovative flood forecasting and warning services.

*Acknowledgements.* The authors would like to express their gratitude to François Bourgin for his careful reading of the manuscript. They also thank the french Ministry of Environment (DGPR/SCHAPI) for the financial support of this work, the DREAL Auvergne Rhone Alpes (SPC Grand Delta) and the IGN for supplying part of the data, and the OHM-CV observatory and the HyMeX program for their help in accessing

15  the rainfall data. OHM-CV (http://www.ohmcv.fr) is an observation service supported by the Institut National des Sciences de l'Univers, section Surface et Interfaces Continentales and the Observatoire des Sciences de l'Univers de Grenoble. OHM-CV is a key observation system of the HyMeX program (http://www.hymex.org/). The radar and raingauge datasets were provided by Météo-France, the SPC Grand Delta and Electricité de France.



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




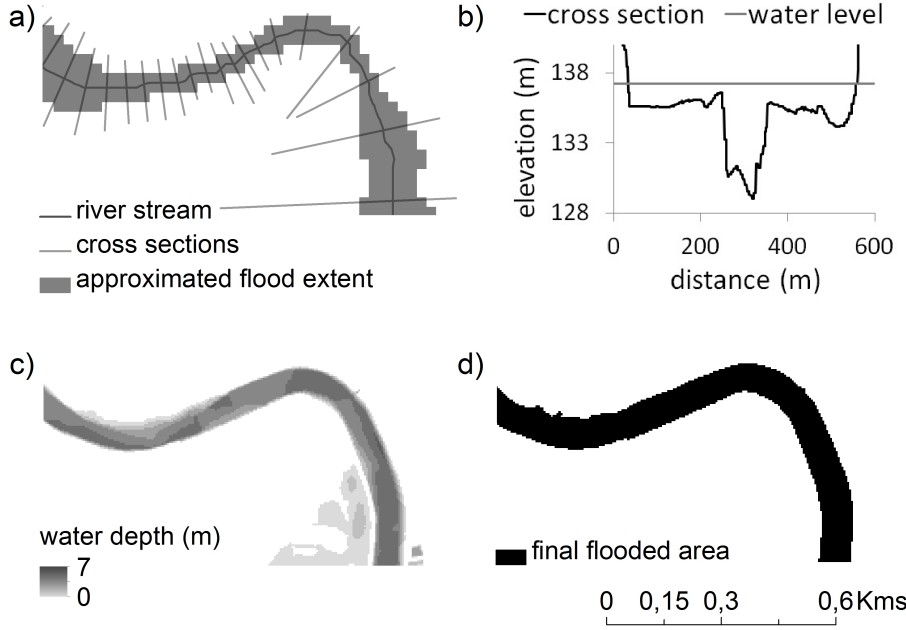

**Figure 1.** Overall principle of the computation of flood maps based on CartinoPC software: a) input information (position of river streams and approximated extent of flooded area) and position of cross-sections, b) example of one cross-section including the computed water level (1-D hydraulic model), c) map of flooded areas and water depths obtained after post-treatment, d) final map of flooded areas after removal of disconnected areas.

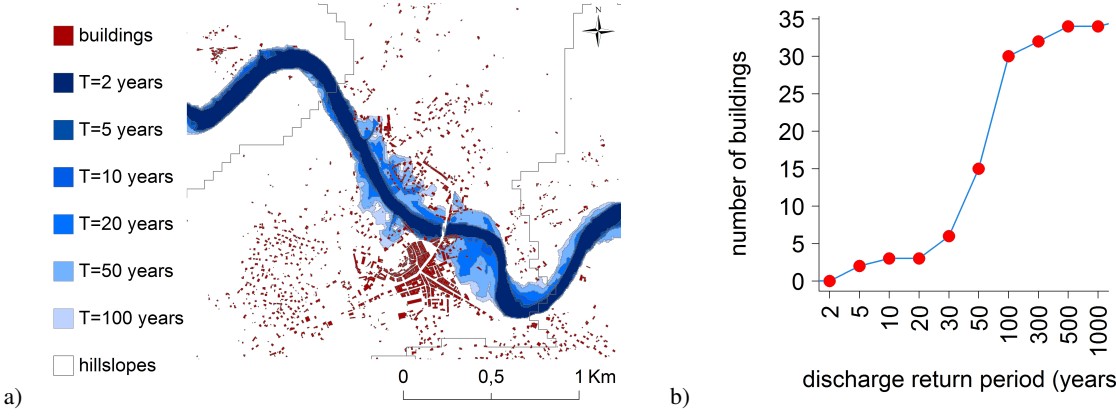

**Figure 2.** Illustration of the implementation of the impacts model on one river reach: a) Catalog of flooded areas, b) interpolated discharge-impacts relation



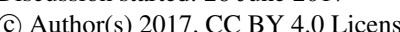



**Figure 3.** Location of the case studies: a) APSFR of Alès territory in the Gardon and Cèze watersheds, b) region of Draguignan in the Argens watershed, and river networks considered in the impacts models (5 $km^2$ upstream catchment surface): c) APSFR of Alès, d) region of Draguignan

**Table 1.** Characteristics of the river networks considerered in the two case studies (extracted from SYRAH database)

| Case study | | bed slope (percent) | river bed width (m) | floodplain width (m) |
|---|---|---|---|---|
| | | ave / min-max | ave / min-max | ave / min-max |
| Alès | main network | 0.48 / 0.22 - 1 | 37 / 16 - 84 | 470 / 120 - 1670 |
| | secondary network | 3.41 / 0.17 - 20 | 7 / 2 - 34 | 430 / 60 - 3130 |
| Draguignan | main network | / 0.06 - 1.2 | 25 / 5.5 - 41 | 730 / 130 - 3200 |
| | secondary network | / 0.03 - 21 | 4 / 1 - 11 | 380 / 3 - 4800 |





**Figure 4.** Examples of comparison of cross-sections and water levels at three stream gauges: a) Mialet station, b) Banne station, c) Alès station, d) Alès station: position of cross-sections and longitudinal profile of river bed (B) and water levels (H)




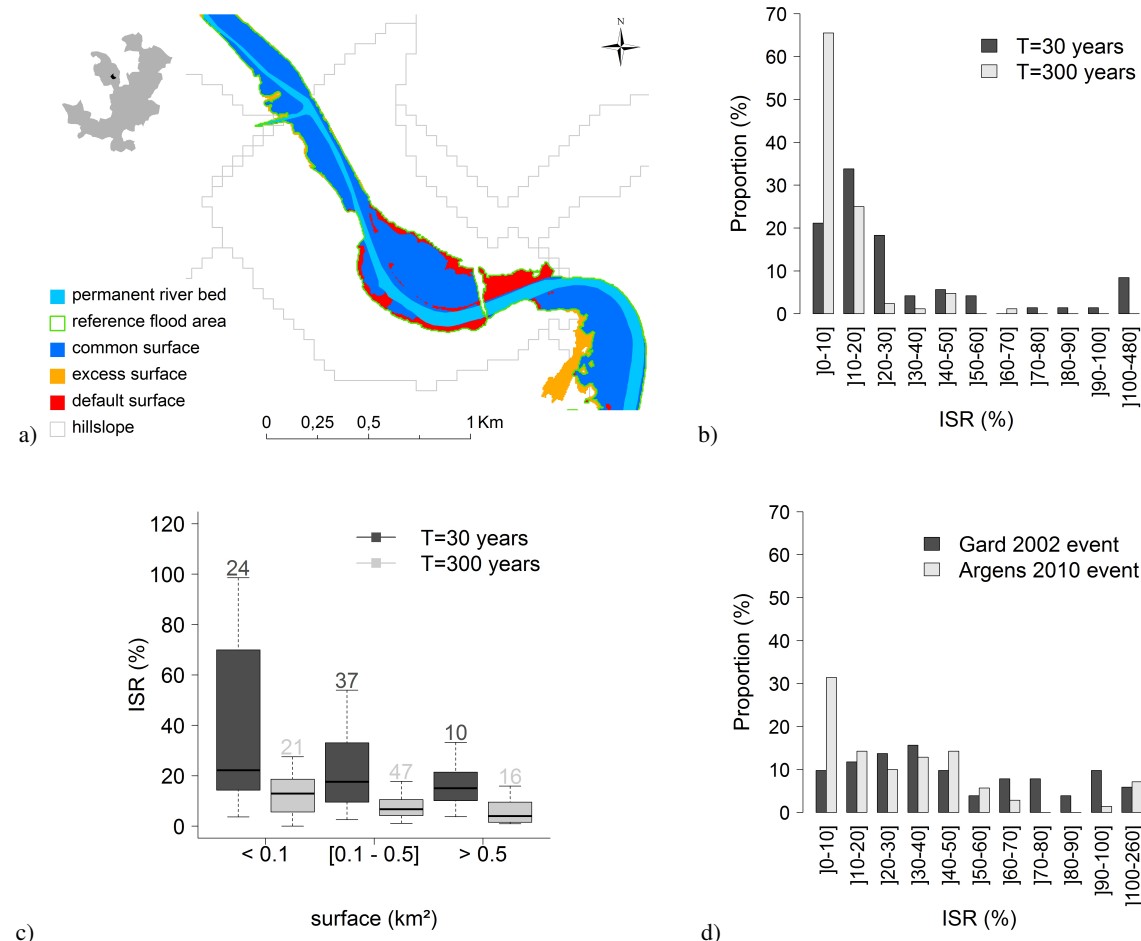

**Figure 5.** Comparison between estimated and reference flooded areas: a) definition of common surface ($S_c$), excess surface ($S_e$), and default surface ($S_d$) for the computation of ISR ratios, b) and c) distributions of ISR scores for the 30-year and 300-year discharge quantiles on the APSFR of Alès, and d) distributions of ISR scores for the full simulation chain for the Alès 2002 and Draguignan 2010 events

**Table 2.** Mean claim/policies and policies/buildings ratios computed inside the floodplains for the september 2002 and june 2010 events. The claims and policies are those selected in the CCR database according to the rules described in section 3.6. The buildings considered are buildings of less than 7m height according to the IGN BD Topo database.

|  | Mean claims/policies ratio | Mean policies/buildings ratio | Global claim/buildings ratio |
|---|---|---|---|
| September 2002 | 0.37 | 0.17 | 0.06 |
| June 2010 | 0.43 | 0.21 | 0.09 |



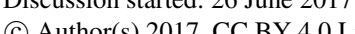



**Figure 6.** Examples of comparison between observed and estimated flood extents, and related TSI values: a) Alès 2002 event for the Amous (left) and Avène (right) rivers, and b) Draguignan 2010 event for the Nartuby (left) and Florieye/Argens (right) rivers





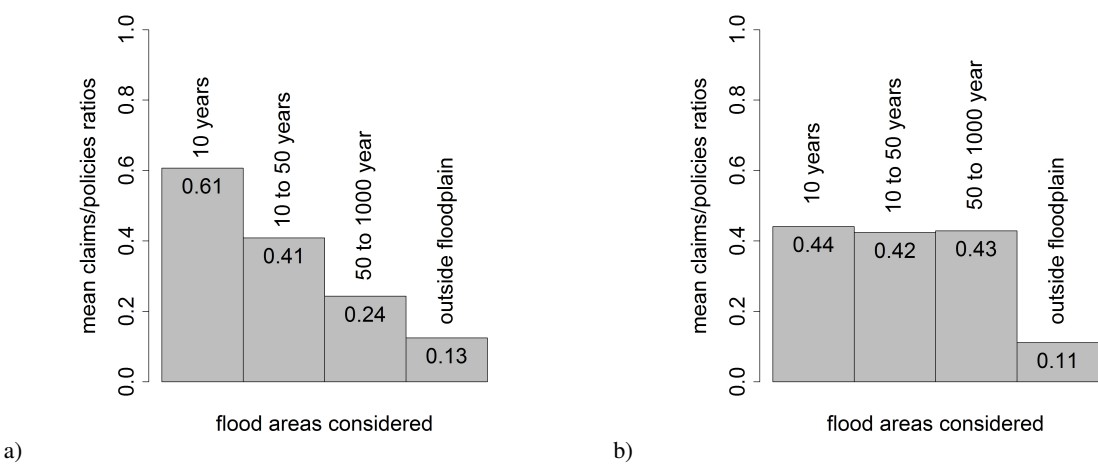

**Figure 7.** Spatial distribution of mean claim/policies ratios, inside and outside the floodplains: a) Alès 2002 event, b) Draguignan 2010 event

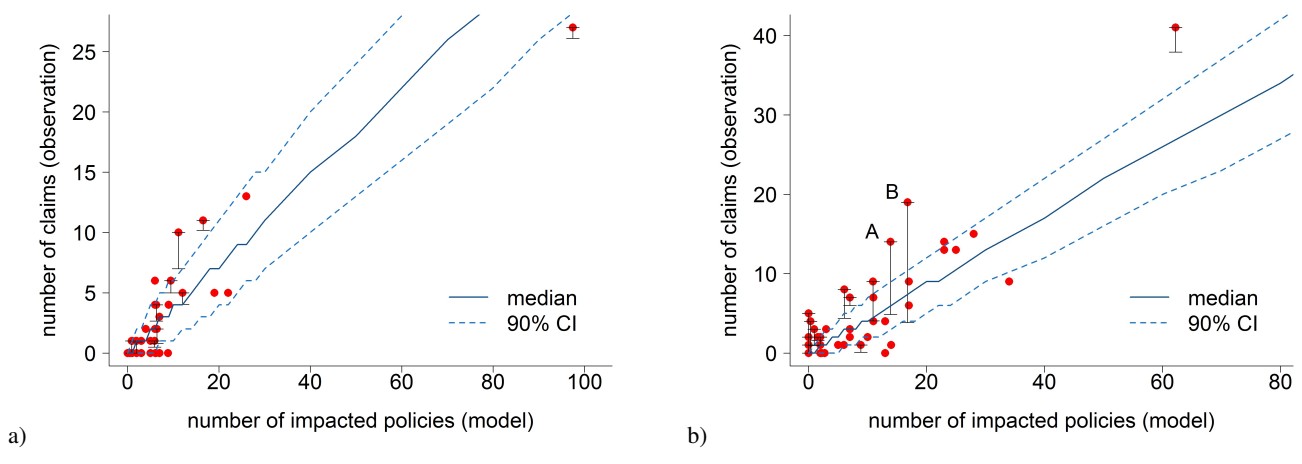

**Figure 8.** Comparison of estimated number policies affected by the flood (model), and associated number of claims for each river reach: a) Alès 2002 event, b) Draguignan 2010 event. Blue lines correspond to the mean claim/policies binomial relation and associated 90% confidence interval, vertical bars represent uncertainty on the number of claims related to river flooding





**Figure 9.** Maps of the peak discharges (return periods) and related impacts (number of flooded policies) simulated by the model, and of the number of claims extracted from the CCR database: a) peak discharges, Alès 2002, b) peak discharges, Draguignan 2010, c) estimated impacts, Alès 2002, d) estimated impacts, Draguignan 2010, e) claims, Alès 2002, f) claims, Draguignan 2010.