# Peer review of "The challenge of forecasting impacts of flash floods: test of a simplified hydraulic approach and validation based on insurance claim data"

_Hydrology and Earth System Sciences, 2017_

## Referee Comment (RC1) · Anonymous Referee #1 · 1 Aug 2017

General comments:

This paper deals with a very interesting and challanging issue: how to forecast the impacts of flash flood (FF)? Indeed, usually, flood forecasting related papers only deal separately with one issue: meteorology, hydrology, social sciences. This paper proposes a common framework and presents a FF forecasting impact method, based on a 2-step chain: 1/ pre-determination of flooded areas allowing to define "potential" impacts and 2/ a real-time rainfall runoff model used to choose the relevent predetermined flooded area. This two-step approach is convincing. It is simple enough

to be run in real-time. On the other hand, the needed simplifications (for instance in the hydraulic model) are carefully identified and analyzed on very well documented two case studies. Finally a "validation" is presented, based on insurance claim data.

Considering the high quality of this paper, I would recommend its publication with only a minor revision.

My main suggestions (detailed in the following remarks) are:

- Some details in the methodology need to be better explained;

- I am not sure that we could talk about "validation based on insurance claim data" as stated in the title, since as it is well explained in the paper, the ratio "nb of claim"/ "nb of flooded policy" does not reach 100% and need here to be event-based calculated using observation data (see my last remarks).

Detailed comments

1. Page 3, Line 19: reference not suitable : this authors are talking about the FFG, which is typically dedicated to ungauged catchment. Find some references dealing only with gauged systems

2. Page 3, Line 20: Figure 1 and text are not totally coherent. The text mention 3 steps and the figure has 4 illustrations. After reading, it seems that illustrations a and b are related to step one and illustration c and d are relatated to step 3. Step 2 is not illustrated (1-D longitudinal water profil computation). I would suggest to make clearly appear the 3 steps on the figure.

3. Page 3, Line 23: Too few information is provided for step 3. How exactly do you move from the 1D longitudinal model to a 2D representation of the flooded area? Furthermore, from the Fig1 legend, it is understood that a (manual?) post treatment is applied to remove some disconnected area, not clearly mentioned in the text.

4. Page 4, Line 13: maybe detail and regroup here all the manual corrections (including

the removal of disconnected area if it is manually done, (see previous rq), in order to insist on problem of using an 100% automated method, and on the consequent amount of work you have done to deal with this issue in order to obtain coherent inundation areas.

5. Page 4, Line 15: The mentioned return periods are not coherent with the Fig2: 6 return periods on a), 10 on b). Furthermore, in the text, latter you also mentioned 10 flooded areas (page 5, line 18). I imagine the 10 return periods you analyzed are those of fig2b ie : 2, 5, 10, 20, 30, 50, 100, 300 (not 200), 200, 1000)?

6. Page 4, Line 15: Do you have just a upstream flow input or also lateral flow inputs? If yes, how this lateral input is estimated and injected in the hydraulic model?

7. Page 4, Line 17: Which limit condition do you apply at the very last downstream reach ? How do you determine the length of each reach?

8. Page 4 Lines 21-24: this point maybe need to be explained more explicitely (gives some partial answer to my rq 3).

9. Page 4 section 2.2: It seems to me that this section focuses more on the limits of the method than on the "catalogue of flood maps" as indicated by the title. I would change the name of this section to clearly indicates that you focus on the limits of the auto-mated procedure presented in 2.1. I would also regroup in this section all the content related to manual procedures earlier mentioned in section 2.1. In this case, lines 15-19 should be put in 2.1. . . .or maybe in 2.3 which is a little too short in comparison with the other sections. (nota: this change in your plan is only a suggestion)

10. Page 5 Line 18: "10 flooded" area ? see rq 5.

11. Page 6 Line 3-4: what spatial and temporal resolution were finally adopted for the rainfall-runoff model in your case studies ? (maybe to be mentioned later, but I didn t find the info in the rest of the paper)

12. Page 6 Line 6: If I understood, the "reaches" and cross sections defined here

are not the same that in the previous section. How are they determined? Are those mentioned in table 1?

13. Page 6 Line 13: "the width of river reaches is the main parameter controlling the transfer function and is estimated based on the Strahler order of river reaches" but previously you wrote Line 6 : "The width of the cross-section varies with return period of the discharge". Could you please clarify exactly how widths are calculated (same rq as 12)

14. Page 8 Line 17: temporal and spatial resolution?

15. Page 8 Line 20: may be change the title by:Reference flood maps obtained from previous studies" (just a suggestion)

16. Page 10, Line 7: There is a jump in the figures numbers. Maybe renumber the figure 7 in figure 4

17. Page 10, Line 32: "It was worse testing if it could provide a number of private houses affected by the floods for each river reach to be compared to the outputs of the proposed forecasting chain." Worth instead of worse? Please reformulate. What do you exactly mean?

18. Page 11 Line 12: do you know how hydrometric services extrapolate these rating curves? Using hydraulic consideration?

19. Page 11 Line 32: replace 'form" by "from"

20. Page 12 Line 21: "This explain why..." Please explain why.

21. Page 13 Line 31: How is your rainfall-runoff model initialized (in particular the initial soil moisture conditions)? Don t you think that can also be a (important) source a uncertainty?

22. Page 13 Line 31: "forecasted peak value" => Maybe replace "forecasted peak value" by "simulated peak value" in all the text since, the RR model is run in a "simulation" not a "forecasting" mode (using QPE, not QPF).

23. Page 14 line 2: In figure 6 legend replace TSI by ISR

24. Page 14 line 12: maybe change "forecasted peak by "simulated peak".

25. Page 14 line 26:"Clearly, the number [. . .] has a random component". I agree, but your random binomial process take into account 'all' the errors of the damage database.

26. Page 14 line 29: Maybe indicate explicitly that you choose p=0.37 and p=0.43 from table 2, so knowing the number of "observed" claim divided by the number of policies effectively flooded (ie using the observed flooded area). Do I correctly understand? Or do you divide by the number of policy flooded according to your model (ie using the modelled flooded area)? In this last case, it is not a validation since p seems to me a kind of last adjustment parameter which unbiased the impact model. Please clarify.

27. Figure 8: But I maybe missed one point. Why can we see some "horizontal" steps into the claim/policies relation (blue). How did you exactly plot these blue curves?

28. Page 15 line 4-5: I don't understand. I thought you used the 'observed' flooded area to calculate the total number of policy (see rq 26). Please clarify.

29. Page 16: I globally agree with the conclusion. But does your paper also suggest that this insurance data are not suitable for a 'real' validation of the impacts (if you mean that "impact" = "nb of claim"), since you need to know the ratio "number of claim"/ "number of impacted policy". . . Maybe further researches are also needed in this direction? How to better estimate this ratio (by removing the other errors you mentioned from the database)? And then how to "forecast" it (is there some national coherences?)?

---

## Referee Comment (RC2) · Anonymous Referee #2 · 13 Aug 2017

General comments:

The idea presented in this paper is interesting and potentially very useful. The paper is well structured and well written. The authors present a novel procedure aimed at computing a series of flood extent maps on a dense stream network and directly evaluating the possible associated impacts. The proposed approach consists of an integrated forecasting chain that combines a one-dimensional simplified hydraulic model and a distributed rainfall-runoff model for the simulation of discharges over the stream networks, and it has been tested on insurance claim data. Potential limitations and critical

issues in the implementation of the proposed methodology are also well documented and discussed throughout the paper. In my opinion, this paper can be considered for publication in the present form, after some minor points will be taken into account. Two main issues and some minor comments are listed below.

1. In section 2, the automatic implementation of 1-D hydraulic models is described; however, the description of the third step is too short and should be completed with additional information. Moreover, from figure 1 it seems that four steps (a-b) are required for obtaining the final map of flooded areas, that is not clear from the text. Finally, the choice of a fixed roughness coefficient is mentioned: a suitable reference should be inserted here. How is this value computed? And what are the possible consequences of keeping it fixed?

2. If I understood well, the application of the method exclusively focuses on the flood peak as the variable of interest. However, it should be stressed that other variables (like, e.g., the flood volume and the flood duration) may play a significant role to the study of extreme flood events, and that the dependence among such variables can seriously influences the estimates of flood magnitudes (see, e.g., Salvadori, G., De Michele, C., and Durante, F.: On the return period and design in a multivariate framework, Hydrol. Earth Syst. Sci., 15, 3293-3305, 2011).

Minor comments

- Page 5 - lines 16-17: You mentioned 10 flooded areas, but I could not find them in figure 2.a. Please also check the return periods reported on the x-axis in figure 2.b which differ from the one mentioned at the beginning of section 2.2.

- Page 6 - lines 23-24: Here, it may be appropriate to clarify that "the continuous discharge-impacts relations" are the continuous curves obtained by linear interpolation that express the relations between discharge return periods and number of impacted insurance policies.

- Page 8 - line 8: "September" and "June".

- Page 8 - line 15: I suggest to specify here that the rating curves are graphs of discharge versus stage for a given point on a stream, and maybe add a comment on how such curves are extrapolated in your work.

- Page 10 - lines 32-33: The sentence "It was worse testing if it could provide a number of private houses affected by the floods for each river reach to be compared to the outputs of the proposed forecasting chain" is not clear. Please, reformulate.

- Page 12 - lines 7: replace "figure 4d" by "figure 4.d". Please check the cross-references throughout the paper.

- Page 12 - line 33: A synthetic incoherent surface ratio is here introduced. I suggest to add a comment on such quantity and/or an appropriate reference.

- Page 13 - lines 22-23-24: Remove the space before the semicolon.

- Please check the punctuation of the figures' captions.

————————————————————

---

## Author Comment (AC1) · 5 Sep 2017

General comments:

This paper deals with a very interesting and challanging issue: how to forecast the impacts of flash flood (FF)? Indeed, usually, flood forecasting related papers only deal separately with one issue: meteorology, hydrology, social sciences. This paper pro poses a common framework and presents a FF forecasting impact method, based on a 2-step chain: 1/ pre-determination of flooded areas allowing to define "potential" impacts and 2/ a real-time rainfall runoff model used to choose the relevent predetermined flooded area. This two-step approach is convincing. It is simple enough to be run in real-time. On the other hand, the needed simplifications (for instance in the hydraulic model) are carefully identified and analyzed on very well documented two case studies. Finally a "validation" is presented, based on insurance claim data. Considering the high quality of this paper, I would recommend its publication with only a minor revision.

My main suggestions (detailed in the following remarks) are:
- Some details in the methodology need to be better explained;
- I am not sure that we could talk about "validation based on insurance claim data" as
stated in the title, since as it is well explained in the paper, the ratio "nb of claim"/ "nb
of flooded policy" does not reach 100% and need here to be event-based calculated
using observation data (see my last remarks).

We thank referee n°1 for the careful reading and globally positive appreciation of our manuscript, and for having raised these two main weaknesses. We explain hereafter how we plan to address these important points (see our answers to detailed comments).

Detailed comments
1. Page 3, Line 19: reference not suitable : this authors are talking about the FFG, which is typically dedicated to ungauged catchment. Find some references dealing only with gauged systems.
OK, we propose here to keep these references but to reformulate the presentation of flash flood guidances, which are to our knowledge among the first approaches really devoted to flash flood warning: "The first approaches developed, namely the flash flood guidances, were based on a preliminary analysis of rainfall volumes generating bankfull flow, for several durations and catchment initial moisture conditions (Georgakakos, 2006; Norbiato et al., 2008). More recent approaches aimed to directly forecast peak discharges at ungauged locations based on distributed hydrological models and radar based quantitative precipitation estimates or nowcasts ( … ".

2. Page 3, Line 20: Figure 1 and text are not totally coherent. The text mention 3 steps and the figure has 4 illustrations. After reading, it seems that illustrations a and b are related to step one and illustration c and d are relatated to step 3. Step 2 is not illustrated (1-D longitudinal water profil computation). I would suggest to make clearly appear the 3 steps on the figure.
3. Page 3, Line 23: Too few information is provided for step 3. How exactly do you move from the 1D longitudinal model to a 2D representation of the flooded area? Furthermore, from the Fig1 legend, it is understood that a (manual?) post treatment is applied to remove some disconnected area, not clearly mentioned in the text.
Actually figures 1.a to 1.c correspond to the three computation steps mentioned in section 2.1 (l.20-23), and figure 1.d to the additional post-treatments described in section 2.2 (p.4 l.21-24 and l.32-33). We suggest here to group the description of all these treatments in section 2.1 (see answer to point 4 below), and to reformulate the text for an explicit reference to the appropriate figure for each step described. The caption of figure 1 will also be changed for an explicit reference to the computation steps described in the text.

4. Page 4, Line 13: maybe detail and regroup here all the manual corrections (including the removal of disconnected area if it is manually done, (see previous rq), in order to insist on

problem of using an 100% automated method, and on the consequent amount of work you have done to deal with this issue in order to obtain coherent inundation areas.

Ok the description of additional treatments (corresponding to figure 1.d), will be moved from section 2.2 and grouped here (i.e. at the end of section 2.1). We will also mention here that these additional treatments could be applied automatically (even if it was done manually here).

5. Page 4, Line 15: The mentioned return periods are not coherent with the Fig2: 6 return periods on a), 10 on b). Furthermore, in the text, latter you also mentioned 10 flooded areas (page 5, line 18). I imagine the 10 return periods you analyzed are those of fig2b ie : 2, 5, 10, 20, 30, 50, 100, 300 (not 200), 200, 1000)?

The 10 return periods are indeed the following: 2,5,10,20,30,50,100,300,500,1000. The list in the text has therefore to be corrected. Figure 2.a includes only one part of these computation results: the T=30, 300, 500, and 1000 years maps are not included. We propose to modify this figure: see the joint updated version including a comprehensive representation of the results.

6. Page 4, Line 15: Do you have just a upstream flow input or also lateral flow inputs? If yes, how this lateral input is estimated and injected in the hydraulic model?

Only an upstream flow input was applied here for the computations, since the information on discharge quantiles was available only at these input nodes. Therefore the discharge is assumed to be homogeneous on each river reach. A comment on this important simplification will added in the text. Fortunately, the considered river reaches are not very long (~2 km in average) and the effects of this simplification should be limited. However, if necessary, running the hydraulic computations with lateral inputs would be possible based on a continuous discharge quantiles information along the stream network.

7. Page 4, Line 17: Which limit condition do you apply at the very last downstream reach ? How do you determine the length of each reach?

The limits of each river reach are determined by the output points of the rainfall-runoff model (which correspond here to the outlets of the french watershed database provided by the SCHAPI): this information will be added in the text (with a reference to sections 3.1 and 3.2 providing information on the lengths of the considered reaches). The dowsnstream boundary condition corresponds to the normal depth for a uniform flow regime: this will also be mentioned in the text.

8. Page 4 Lines 21-24: this point maybe need to be explained more explicitly (gives some partial answer to my rq 3).

See above the answers to remarks 3 and 4: this explanation on additional post-treatments will be detailed and moved to the end of section 2.1, with further comments on the possibility of automatization of these treatments.

9. Page 4 section 2.2: It seems to me that this section focuses more on the limits of the method than on the "catalogue of flood maps" as indicated by the title. I would change the name of this section to clearly indicates that you focus on the limits of the automated procedure presented in 2.1. I would also regroup in this section all the content related to manual procedures earlier mentioned in section 2.1. In this case, lines 15-19 should be put in 2.1. or maybe in 2.3 which is a little too short in comparison with the other sections. (nota: this change in your plan is only a suggestion)

Our proposal is to reorganize the plan in the following way (by grouping sections 2.2 and 2.3):
2.1 Simplified automatic implementation of 1-D steady-state hydraulic models
2.2 Computation of a catalog of flood extent maps and associated impacts model
2.3 Remaining limits of the procedure
2.4 The rainfall-runoff-impacts simulation chain

10. Page 5 Line 18: "10 flooded" area ? see rq 5.

10 flood maps are actually used for the computation of the impacts models. See answer to remark 5.

11. Page 6 Line 3-4: what spatial and temporal resolution were finally adopted for the rainfall-runoff model in your case studies ? (maybe to be mentioned later, but I didn t find the info in the rest of the paper)
Some elements on the spatial resolution (5km² elementary catchment area for the definition of river network integrated in the model) are provided in section 3 but are probably not sufficient. The temporal resolution of the computations is 15 min. A sentence will be added here to detail these aspects.

12. Page 6 Line 6: If I understood, the "reaches" and cross sections defined here are not the same that in the previous section. How are they determined? Are those mentioned in table 1?
The river reaches considered in the hydraulic computations and in the rainfall runoff model are identical: this will be mentioned by adding a sentence on the model resolution (see previous point). However, the geometrical characteristics of the cross sections are simplified and assumed to be rectangular in the rainfall-runoff model, since only the flood wave transit time has to be well anticipated (see next remark for the determination of the cross sections widths).

13. Page 6 Line 13: "the width of river reaches is the main parameter controlling the transfer function and is estimated based on the Strahler order of river reaches" but previously you wrote Line 6 : "The width of the cross-section varies with return period of the discharge". Could you please clarify exactly how widths are calculated (same rq as 12)
The widths are estimated based on a generic formula proposed by Versini (2010): $W=Lo.i^2$, where i is the Strahler order and Lo is an elementary width having 3 different possible values depending the discharge return period. These details are provided in the cited reference (Naulin, 2013), and are not crucial here in our opinion. Therefore we propose to remove the sentence on line 6 and to reformulate line 13 :"the width of the river reaches is the main parameter controlling the transfer function and is estimated based on the Strahler order and discharge return period (for further details see Naulin et al., 2013)"

14. Page 8 Line 17: temporal and spatial resolution?
1 km² of spatial resolution, 5 min of temporal resolution in both cases. However, this does not correspond to the resolutions of rainfall runoff computations (see above).

15. Page 8 Line 20: may be change the title by:Reference flood maps obtained from previous studies" (just a suggestion)
OK, we agree.

16. Page 10, Line 7: There is a jump in the figures numbers. Maybe renumber the figure 7 in figure 4.
OK, this will be done.

17. Page 10, Line 32: "It was worse testing if it could provide a number of private houses affected by the floods for each river reach to be compared to the outputs of the proposed forecasting chain." Worth instead of worse? Please reformulate. What do you exactly mean?
We propose a simplification of this sentence: "We tested herein if it could provide a number of private houses affected by the floods to be compared to …"

18. Page 11 Line 12: do you know how hydrometric services extrapolate these rating curves? Using hydraulic consideration?
The extrapolation may be based either on a local hydraulic modelling (Alès and Mialet) or on an expert-analysis of hydraulic control of the streamgauge section (Banne). These important precisions will be added in the text.

19. Page 11 Line 32: replace 'form" by "from"
OK.

20. Page 12 Line 21: "This explain why..." Please explain why.
We propose this reformulation "It is important to note that the low-flows water heights are limited in these Mediterranean rivers. This explains why bathymetric data is not crucial to obtain a relevant estimation of the cross sections shapes: aerial topographic surveys are often sufficient to get an accurate representation of the river beds in Mediterranean regions."

21. Page 13 Line 31: How is your rainfall-runoff model initialized (in particular the initial soil moisture conditions)? Don t you think that can also be a (important) source a uncertainty?
Indeed the initialization of the rainfall runoff model is a crucial aspect which can result in significant bias in the estimation of peak discharges. The model initialization is performed here based on the initial method proposed by USDA, using the 5 days antecedent rainfall: this will be mentioned in section 2.4 with a reference to Naulin (2013) for details. More generally, the rainfall-runoff simulation represents a significant part of the final uncertainty of the whole simulation chain. This is illustrated by figures 5.b and 5.d, and commented in section 4.3.

22. Page 13 Line 31: "forecasted peak value" => Maybe replace "forecasted peak value" by "simulated peak value" in all the text since, the RR model is run in a "simulation" not a "forecasting" mode (using QPE, not QPF).
OK, this will be done.

23. Page 14 line 2: In figure 6 legend replace TSI by ISR
OK.

24. Page 14 line 12: maybe change "forecasted peak by "simulated peak".
OK.

25. Page 14 line 26:"Clearly, the number [...] has a random component". I agree, but your random binomial process take into account 'all' the errors of the damage database.
We propose here to modify the sentence: "Clearly, the number of reported claims per river reach has a random component due both to the claims triggering processes and to the limits of the claim database."

26. Page 14 line 29: Maybe indicate explicitly that you choose p=0.37 and p=0.43 from table 2, so knowing the number of "observed" claim divided by the number of policies effectively flooded (ie using the observed flooded area). Do I correctly understand? Or do you divide by the number of policy flooded according to your model (ie using the modelled flooded area)? In this last case, it is not a validation since p seems to me a kind of last adjustment parameter which unbiased the impact model. Please clarify.
Yes, the ratios should be computed inside the actually observed flood area, and without using the computed flood areas. However, the observed flood area may not be available everywhere. By default, our proposal here is to compute the C/P ratios on the 1000-year flood extent (which differs from the simulated flood area for the considered events). This may lead to slightly underestimate the C/P ratios since some non-flooded areas are incorporated in the estimation of the ratio. A sentence will be added in the text to clarify this point.

27. Figure 8: But I maybe missed one point. Why can we see some "horizontal" steps into the claim/policies relation (blue). How did you exactly plot these blue curves?
The binomial process correspond to a discrete random variable: each realization c of the variable is necessarily an integer (c being the number of claims reported for one river reach). The nonexceedance probabilities F(c) increase by steps in function of c and cannot be exactly equal to 0.05, 0.50, or 0.95. For this reason, the intervals limits were computed here as the lowest discrete c values for which $F(c) > \alpha$, $\alpha \in (0.05, 0.50, 0.95)$. This explains the steps in the curves especially when the number of claims is limited. To avoid this, we propose to estimate the intervals limits by linear interpolation between c and c-1, with $F(c-1) < \alpha < F(c)$, $\alpha \in (0.05, 0.50, 0.95)$. We join the updated figures.

28. Page 15 line 4-5: I don't understand. I thought you used the 'observed' flooded area to calculate the total number of policy (see rq 26). Please clarify.
Provided that the observed flood area is not available everywhere, our choice was again here to estimate the reference number of claims based on the 1000-year flood envelop (i.e. without using the simulated flood area, see answer to remark 26). This time, this choice may result in a significant overestimation of the number of claims observed in the actual flood area.

29. Page 16: I globally agree with the conclusion. But does your paper also suggest that this insurance data are not suitable for a 'real' validation of the impacts (if you mean that "impact" = "nb of claim"), since you need to know the ratio "number of claim"/ "number of impacted policy"... Maybe further researches are also needed in this direction?
How to better estimate this ratio (by removing the other errors you mentioned from the database)? And then how to "forecast" it (is there some national coherences?)?
Clearly the validation process could still be improved. First, by a better identification of claims really related to river flooding in the claim database, which is unfortunately out of reach at this time. A predictive estimation of the claim ratios would also certainly enhance the validation process. Such estimations are usually obtained by insurance companies based on mean relations between claims ratios and inundation water levels, generally established at a large scale. Therefore, a further research step could be to use the water levels from our 1-D hydraulic computations to directly estimate the claim ratios. This consideration will be added in the conclusion.

---

## Author Comment (AC2) · 5 Sep 2017

General comments:

The idea presented in this paper is interesting and potentially very useful. The paper is well structured and well written. The authors present a novel procedure aimed at computing a series of flood extent maps on a dense stream network and directly evaluating the possible associated impacts. The proposed approach consists of an integrated forecasting chain that combines a one-dimensional simplified hydraulic model and a distributed rainfall-runoff model for the simulation of discharges over the stream networks, and it has been tested on insurance claim data. Potential limitations and critical issues in the implementation of the proposed methodology are also well documented and discussed throughout the paper. In my opinion, this paper can be considered for publication in the present form, after some minor points will be taken into account. Two main issues and some minor comments are listed below.

At first we would like to thank the referee n°2 for this positive appraisal of our manuscript, and for the detailed comments and suggestions provided. Our answers are detailed below.

1.  In section 2, the automatic implementation of 1-D hydraulic models is described; however, the description of the third step is too short and should be completed with additional information. Moreover, from figure 1 it seems that four steps (a-b) are required for obtaining the final map of flooded areas, that is not clear from the text. Finally, the choice of a fixed roughness coefficient is mentioned: a suitable reference should be inserted here. How is this value computed? And what are the possible consequences of keeping it fixed?

This remark is close to another one formulated by referee n°1. Actually figure 1 illustrates both the three computation steps mentioned in section 2.1 (p.3 l.20-23), and the additional post-treatments described in section 2.2 (p.4 l.21-24 and l.32-33). We propose to group the description of all these treatments in section 2.1 and to reformulate the text for an explicit reference to the appropriate figures. In the same time the description of the third step (p3 l.23) will be detailed as follows:" the estimated water levels are interpolated between successive cross sections and compared to the DTM elevations to compute the flood extent and water depth maps (figure 1.c) ".

The roughness coefficient was fixed according Lumbroso et al. (2012) who showed the necessity to limit the roughness values to keep reasonable values of flow velocities. This reference will be added.

The consequence may be a systematic overestimation of the water levels. But according to the results obtained in section 4, it seems it is not the case here.

2. If I understood well, the application of the method exclusively focuses on the flood peak as the variable of interest. However, it should be stressed that other variables (like, e.g., the flood volume and the flood duration) may play a significant role to the study of extreme flood events, and that the dependence among such variables can seriously influences the estimates of flood magnitudes (see, e.g., Salvadori, G., De Michele, C., and Durante, F.: On the return period and design in a multivariate framework, Hydrol. Earth Syst. Sci., 15, 3293-3305, 2011).

Indeed the flood duration and volume may largely influence the flood extents, particularly in the case of large rivers with wide floodplains. These variables could be accounted for by using synthetic hydrographs and unsteady hydraulic computations. However this procedure would largely increase the computation times, and would require comparing the reference synthetic hydrographs to the actually simulated ones for the application of the whole simulation chain. In the particular context of flash floods (very fast evolution and limited lead times), we considered here it was preferable to simplify the procedure, through the assumption that the flood extent is mainly influenced by the peak discharge. This enables to conduct the hydraulic computations in steady-state regime. This assumption is clearly presented as ones of the limits of the procedure (see section 2.2, p.5 l.10). It nevertheless remains reasonable in our opinion since we are here working on small upstream watersheds, in relatively hilly areas. A comment will be added in section 2.2 about the reasons for this choice.

Minor comments

- Page 5 - lines 16-17: You mentioned 10 flooded areas, but I could not find them in figure 2.a. Please also check the return periods reported on the x-axis in figure 2.b which differ from the one mentioned at the beginning of section 2.2.

Yes indeed the text includes here an error and has to be corrected. The 10 return periods used are actually the following: 2,5,10,20,30,50,100,300,500,1000. Figure 2.a presents only one part of the computation results. An updated version is proposed (see joint file) including all computed flood extents.

- Page 6 - lines 23-24: Here, it may be appropriate to clarify that "the continuous discharge-impacts relations" are the continuous curves obtained by linear interpolation that express the relations between discharge return periods and number of impacted insurance policies.

Yes the relations are obtained by linear interpolation. We propose here to add a reference to section 2.3 and figure 2 which provide details on this important methodological aspect.

- Page 8 - line 8: "September" and "June".

Ok this will be corrected (page 11 line 8).

- Page 8 - line 15: I suggest to specify here that the rating curves are graphs of discharge versus stage for a given point on a stream, and maybe add a comment on how such curves are extrapolated in your work.

The extrapolation may be based either on a local hydraulic modelling (Mialet and Alès) or on an expert-analysis of the hydraulic control of the streamgauge section (Banne). This precision will be added in section 4.1.

- Page 10 - lines 32-33: The sentence "It was worse testing if it could provide a number of private houses affected by the floods for each river reach to be compared to the outputs of the proposed forecasting chain" is not clear. Please, reformulate.

We propose a simplification of this sentence: "We tested herein if it could provide a number of private houses affected by the floods to be compared to …"

- Page 12 - lines 7: replace "figure 4d" by "figure 4.d". Please check the crossreferences throughout the paper.

OK, to be corrected.

- Page 12 - line 33: A synthetic incoherent surface ratio is here introduced. I suggest to add a comment on such quantity and/or an appropriate reference.

We propose to add the following comment: "A synthetic incoherent surface ratio (ISR) is computed (eq. 1). It represents the extent of excess and default surfaces, expressed as a proportion of the reference surface".

- Page 13 - lines 22-23-24: Remove the space before the semicolon.

OK.

- Please check the punctuation of the figures' captions.

OK.

---

## Referee Comment (RC3) · Anonymous Referee #3 · 8 Sep 2017

**MAJOR COMMENTS**

The present article shows a novel methodology for the estimation of the flash flood impacts using a hydraulic model and a rainfall-runoff model. The article is well written and structured which makes it very understandable. The figures are pretty illustrative and are well explained in the text. Some methodology aspects must be better explained in the text since it is a relevant section for this article and some processes are not mentioned in depth (rating curves, river reaches, better explanation of the models operation, etc). The validation of the impact model with insurance data gives an extra

and innovative point in the article, showing the importance of this data and all the information it can provide. From my point of view, this article is ready for publication, with some minor changes:

SPECIFIC REMARKS

1. Page 3, Line 28: consider using the same punctuation throughout the text. For instance, dots for decimal numbers (n=0.05). I suggest to add some reference explaining why it is used this specific roughness coefficient.

2. Page 4, Line 33: take care with the citation of the figures, it is different throughout the text (i.e. figure 1.d instead of figure 1d).

3. Consider using always the same English spelling (UK or US). For example, in the Figure 2, the word "catalog" is used, however in the text is used "catalogue". The same with the words "modelled" and "modeled".

4. Why "km2" are the only units that are in italics? I suggest putting all them in the same way.

5. Page 7, Line 12: in this section (3.1) the meaning of "river reach" is explained for the first time. Consider explaining it before.

6. Page 8, Line 2: "altimetric" instead of "altmetric".

7. Page 9, Line 19: it is said in the text that is only used private houses, mostly individual houses (>7m height). What about public or commercial buildings? Does the CCR cover them?

8. Write the meaning of all the acronyms appearing in the text for the first time. For example IGN RE (page 8, line 1) or QPEs (page 8, line 17)

9. Consider citing internet sites, instead of including the wrl in the text.

10. Please change the order of the Table 2, since it is mentioned before Figures 4 and

5. The same case with Figure 7, it can't be mentioned in the text before Figures 4, 5, and 6.

11. Consider including more information about the rivers of the case studies, like the average discharge and the maximum peak discharge of both flood events in one of the stream gauges shown in the figure 3.

12. Figure 6: "ISR" instead of "TSI". Which modelled value is used for the ISR estimation? The upper or the lower bound? Why the ISR values are estimated just in one of the case studies (Draguignan 2010)?

13. Page 10, line 32: I don't understand the sentence "It was worse testing if it could provide a number of private houses affected by the floods for each river reach to be compared to the outputs of the proposed forecasting chain".

14. Page 13, line 5 and 7: "ISR" instead of "IRS".

15. Page 15, last paragraph: Figure 9 is wrong mentioned in the text.

16. References: change the order of "Gourley et al." references, since the newest one must be placed after the oldest one.

17. Figures:

- Use always the same units, "km" instead of "kms" (International System)

- Take care with the punctuation of the decimal numbers of the figures.

- The position of the "a); b); c); d)" within the figures must be always the same. Change it in the Figure 1.

- All the captions must have the same format.

---

## Author Comment (AC3) · 20 Sep 2017

MAJOR COMMENTS

The present article shows a novel methodology for the estimation of the flash flood impacts using a hydraulic model and a rainfall-runoff model. The article is well written and structured which makes it very understandable.  The figures are pretty illustrative and are well explained in the text.  Some methodology aspects must be better explained in the text since it is a relevant section for this article and some processes are not mentioned in depth (rating curves, river reaches, better explanation of the models operation, etc).  The validation of the impact model with insurance data gives an extra and innovative point in the article, showing the importance of this data and all the information it can provide.  From my point of view, this article is ready for publication, with some minor changes:

We thank referee n°3 for this detailed review of our manuscript. The suggestions formulated largely meet the remarks from the two other referees. We detail hereafter our answers to each of these specific points:

SPECIFIC REMARKS

1.  Page 3, Line 28:  consider using the same punctuation throughout the text.  For instance, dots for decimal numbers (n=0.05). I suggest to add some reference explaining why it is used this specific roughness coefficient.

This lack of reference was also pointed by referee n°2. Actually the roughness coefficient was fixed according to Lumbroso et al. (2012) who showed the necessity to limit the roughness values to keep reasonable values of flow velocities. This reference will be added.

2.  Page 4, Line 33:  take care with the citation of the figures, it is different throughout the text (i.e. figure 1.d instead of figure 1d).

Ok, this will be checked and corrected as necessary.

3.  Consider using always the same English spelling (UK or US). For example, in the Figure 2, the word "catalog" is used, however in the text is used "catalogue". The same with the words "modelled" and "modeled".

Ok, we will be check this and adopt the same English.

4.  Why "km2" are the only units that are in italics?  I suggest putting all them in the same way.

Ok, we agree all units should be typed in the same way, this will be done.

5. Page 7, Line 12: in this section (3.1) the meaning of "river reach" is explained for the first time. Consider explaining it before.

We propose to move this explanation to the introduction section, since the term river reach is used in this section for the first time.

6. Page 8, Line 2: "altimetric" instead of "altmetric".

Ok, this will be corrected.

7. Page 9, Line 19: it is said in the text that is only used private houses, mostly individual houses (>7m height). What about public or commercial buildings? Does the CCR cover them?

Public buildings are only partly covered, and commercial buildings are generally covered. However we decided to exclude this information, since the addresses of insurance policies often do not correspond to the location of the insured buildings in these cases. Therefore the real location of the damaged buildings cannot be determined accurately.

8. Write the meaning of all the acronyms appearing in the text for the first time. For example IGN RE (page 8, line 1) or QPEs (page 8, line 17)

Ok the meanings will be added at the first apparition.

9. Consider citing internet sites, instead of including the wrl in the text.

The URLs of the different institutions will be moved in the acknowledgements sections.

10. Please change the order of the Table 2, since it is mentioned before Figures 4 and 5. The same case with Figure 7, it can't be mentioned in the text before Figures 4, 5, and 6.

OK the order of tables and figures will be modified to better follow the citations in the text.

11. Consider including more information about the rivers of the case studies, like the average discharge and the maximum peak discharge of both flood events in one of the stream gauges shown in the figure 3.

We propose to add some peak discharge values estimated for both events at several points of the considered river networks. However, since almost all the stations were damaged during the floods, the information on mean discharges is not available.

12. Figure 6: "ISR" instead of "TSI". Which modelled value is used for the ISR estimation? The upper or the lower bound? Why the ISR values are estimated just in one of the case studies (Draguignan 2010)?

OK the acronym will be replaced. The lower bounds have been used for the computation of the ISR values (this will be mentioned). The ISR values are presented on one case study to preserve the readability of the other figures showing the comparison between observed and simulated flood maps (lower and upper bound). We propose to keep this, since the ISR ratios computed on the other case study do not show significantly different features.

13. Page 10, line 32: I don't understand the sentence "It was worse testing if it could provide a number of private houses affected by the floods for each river reach to be compared to the outputs of the proposed forecasting chain".

All the referees pointed out the necessary reformulation of this sentence. It will be replaced by: "We tested herein if it could provide a number of private houses affected by the floods to be compared to …"

14. Page 13, line 5 and 7: "ISR" instead of "IRS".

OK. This will be corrected.

15. Page 15, last paragraph: Figure 9 is wrong mentioned in the text.

OK. This will be modified.

16. References: change the order of "Gourley et al." references, since the newest one must be placed after the oldest one.

OK. This will be done.

17. Figures:

- Use always the same units, "km" instead of "kms" (International System)

- Take care with the punctuation of the decimal numbers of the figures.

- The position of the "a); b); c); d)" within the figures must be always the same. Change it in the Figure 1.

- All the captions must have the same format

OK. This will be done.